# Haematological dynamics following treatment of visceral leishmaniasis: a protocol for systematic review and individual participant data (IPD) meta-analysis

Abdalla Munir [iD] ,[1,2] Prabin Dahal [iD] ,[1,2] Rishikesh Kumar,[3] Sauman Singh-Phulgenda [iD] ,[1,2] Niyamat Ali Siddiqui,[3] Caitlin Naylor,[1,2] James Wilson [iD] ,[1,2] Gemma Buck,[1,2] Manju Rahi [iD] ,[4] Fabiana Alves,[5] Paritosh Malaviya,[6] Shyam Sundar,[6] Koert Ritmeijer,[7] Kasia Stepniewska,[1,2] Krishna Pandey [iD] ,[3] Philippe J Guérin,[1,2] Ahmed Musa[8]

For numbered affiliations see end of article.

**Correspondence to**
Dr Prabin Dahal;
prabin.dahal@ndm.ox.ac.uk and Professor Ahmed Musa;
musaam2003@yahoo.co.uk

## ABSTRACT

**Introduction** Visceral leishmaniasis (VL) is a parasitic disease with an estimated 30 000 new cases occurring annually. Despite anaemia being a common haematological manifestation of VL, the evolution of different haematological characteristics following treatment remains poorly understood. An individual participant data meta-analysis (IPD-MA) is planned to characterise the haematological dynamics in patients with VL.

**Methods and analysis** The Infectious Diseases Data Observatory (IDDO) VL data platform is a global repository of IPD from therapeutic studies identified through a systematic search of published literature (PROSPERO registration: CRD42021284622). The platform currently holds datasets from clinical trials standardised to a common data format. Corresponding authors and principal investigators of the studies indexed in the IDDO VL data platform meeting the eligibility criteria for inclusion were invited to be part of the collaborative IPD-MA. Mixed-effects multivariable regression models will be constructed to identify determinants of haematological parameters by taking clustering within study sites into account.

**Ethics and dissemination** This IPD-MA meets the criteria for waiver of ethical review as defined by the Oxford Tropical Research Ethics Committee (OxTREC) granted to IDDO, as the research consists of secondary analysis of existing anonymised data (exempt granted on 29 March 2023, OxTREC REF: IDDO). Ethics approval was granted by the ICMR-Rajendra Memorial Research Institute of Medical Sciences ethics committee (letter no.: RMRI/EC/30/2022) on 4 July 2022. The results of this analysis will be disseminated at conferences, the IDDO website and peer-reviewed publications in open-access journals. The findings of this research will be critically important for control programmes at regional and global levels, policymakers and groups developing new VL treatments.

## STRENGTHS AND LIMITATIONS OF THIS STUDY

⇒ An individual participant data meta-analysis (IPD-MA) is proposed to characterise the evolution of haemoglobin and other haematological parameters during the study period; this will overcome the limitations due to sample size issues.
⇒ The identification of studies has been made through a comprehensive literature search of all published studies since 1980 with a predefined inclusion–exclusion criteria.
⇒ A major strength of this IPD-MA is that it uses IPD hosted at the Infectious Diseases Data Observatory, which has harmonised raw data to a common standard based on extensive consultation with the visceral leishmaniasis research community.
⇒ The retrieval of data from trials published prior to 2000 can be a major challenge.

**PROSPERO registration number** CRD42021284622.

## INTRODUCTION

Anaemia is a common haematological manifestation of visceral leishmaniasis (VL).[1] In patients with VL, anaemia can arise due to one or more associated factors: sequestration and haemolysis of red blood cells in the spleen associated with hypersplenism, bone marrow suppression caused by nutritional deficiencies such as iron, vitamin B12 and folate deficiencies or clotting dysfunction leading to blood loss.[1–5] At the time of clinical diagnosis, haemoglobin levels are often around 70-100 g/L but can be as low as 40 g/L.[1] The severity of anaemia depends on the duration of the clinical illness and can be exacerbated by comorbidities and iron deficiency.[6 7]

After treatment with an antileishmanial drug, haematological improvement generally occurs within a few weeks, with significant recovery expected within 4–6 weeks.[3] The trajectory of haematological recovery may be affected by the patient's age or initial parasite load,[8] but the influence of drug regimen and other patient characteristics remains poorly understood. The haematological safety of the VL treatment remains an important concern.[9] Therefore, characterisation of the haematological profile and identification of drivers associated with haematological recovery can help in optimal case management. The present individual participant data meta-analysis (IPD-MA) aimed to characterise the haematological changes in patients with VL after treatment and understand the role of host, parasite and drug-related characteristics.

## OBJECTIVES

The objectives of this IPD-MA are:
1. To identify the determinants of haemoglobin concentration at enrolment, at initial cure assessment and at definitive cure assessment.
2. To characterise the haemoglobin dynamics following treatment.
3. To identify the determinants of anaemia or severe anaemia at enrolment, at initial cure assessment and at definitive cure assessment.
4. To describe white blood cell (WBC) and platelet dynamics following treatment (data permitting).
5. To identify predictors of blood transfusion during treatment or follow-up (if available).

## METHODS AND ANALYSIS: PATIENTS, INTERVENTIONS AND OUTCOMES

### Elements of the research aim (PICOT)

*Population*: Any patient enrolled in prospective efficacy studies with a confirmed or suspected diagnosis of VL defined by serological and/or parasitological testing.

*Interventions:* Any antileishmanial therapy.

*Comparator:* Not restricted by the use of a comparator drug.

*Outcome:* At least one of the following outcomes was reported: anaemia (or haemoglobin measurements) at enrolment, anaemia and other haematological measurements at any postbaseline time points.

*Time:* Studies published on or after 1980.

### Criteria for study eligibility

Studies in the Infectious Diseases Data Observatory (IDDO) VL data platform[10] will be deemed eligible for the purpose of this IPD-MA if they meet the following criteria:
▶ Prospective efficacy studies on patients with confirmed or suspected VL, using microscopy, serology or molecular methods (ie, clinical diagnosis followed by a confirmatory diagnosis).

▶ Information is available on the treatment regimen, including the drug, dose and duration of the regimen.
▶ Data on anaemia (or haemoglobin or haematocrit concentration) measured at enrolment.

### Desirable criteria
▶ The methodology used for haematological quantification, for example, the name of the device used.
▶ Information regarding whether blood transfusions were required before or after treatment initiation.
▶ Information on the co-administration of iron supplements.

### Criteria for participant eligibility

The minimum information required for participants from each of the identified studies for inclusion in the IPD-MA analysis is listed below:
▶ Details of the antileishmanial treatment(s) administered.
▶ Baseline information on age and gender.
▶ At least one of the haematological outcomes (eg: haemoglobin) is recorded

## Outcomes
### Outcomes and definitions
*Primary outcome*

The primary outcome is the haemoglobin concentration measured at any time period during the treatment and follow-up phases.

*Secondary outcomes*

The following endpoints are identified as secondary:
▶ Anaemia and severe anaemia at baseline, at the time of initial cure assessment and at the time of definitive cure assessment.
▶ WBCs at baseline and at any time during the treatment and follow-up.
▶ Platelet counts at baseline and at any time during the treatment and follow-up.
▶ Requirement of blood transfusion during treatment or follow-up (if available).

Anaemia will be defined following the WHO guidelines.[11] The timing of the initial cure assessment would typically take place within 28 days of treatment completion but this varies slightly across studies; assessments undertaken between 15 and 60 days will be considered as the time of the initial cure assessment. Similarly, the timing of the definitive cure assessment will vary according to the study design and hence assessments made between 5 and 7 months will be considered to be 6 months.

## STATISTICAL METHODS
### Identification of relevant studies using IDDO VL library

We searched all the articles indexed in the open-access IDDO VL clinical trials library (IDDO VL library).[12] The IDDO VL library is continually updated and follows the Preferred Reporting Items for Systematic Reviews and Meta-Analyses (PRISMA) guidelines.[13] The IDDO VL

library indexes publications identified from the following databases: PubMed, Embase, Scopus, Web of Science, Cochrane, clinicaltrials.gov, WHO ICTRP, Global Index Medicus, IMEMR, IMSEAR and LILACS. For this current review, the search includes all clinical trials published between 1 January 1980 and 2 May 2021. Details of the search strategy adopted are described elsewhere.[13] The search details are presented in online supplemental file 1. Studies indexed in the IDDO VL library will be eligible for inclusion in this review if they meet the inclusion and exclusion criteria outlined above. This review is not limited by language.

### Collating IPD: IDDO VL data platform

Principal investigators and the corresponding authors of the eligible studies identified in the IDDO VL library were invited to share IPD. At least two emails were sent out in case of a non-response. Researchers agreeing to the terms and conditions of the submission were invited to upload anonymised IPD to the IDDO repository through a secure web portal.[10] Data on the IDDO VL platform are fully pseudonymised to protect personal information and patient privacy.

### Data management

Raw data from individual studies shared with IDDO are currently being standardised using the Clinical Data Interchange Standards Consortium (CDISC) compliant curation standards.[14] Investigators will be further contacted for validation or clarification, if required, and individual study protocols will be requested. On standardisation, the data is stored in a relational database of several tables containing information on drug regimen, parasitological, clinical and haematological assessments and therapeutic outcomes.

### STATISTICAL METHODS FOR THE ANALYSIS OF PRIMARY AND SECONDARY OUTCOMES

#### Descriptive summary of the studies included

A summary of the included studies will be presented with respect to study location, years of study, study population, duration of follow-up, drug regimen, methodology for diagnosis, supervision of drug administration or treatment adherence (if available), methods and devices used for haematology measurements.

#### Summary of the participants included

A summary of the baseline characteristics of the participants included in the analysis will be presented for each study, by region and overall. The following will be presented: age, weight, parasite grade on enrolment, presence of fever (body temperature >37.5°C), haemoglobin (or haematocrit), anaemia or severe anaemia as defined using the WHO definitions,[11] spleen size, treatment, total mg/kg dose and supervision of drug administration. The number of available patients will be summarised for all variables; proportions will be used for categorical or binary variables and mean and SD (or median and IQR) will be used for continuous variables.

### Analysis of the primary endpoint

Separate linear mixed-effect regression models will be undertaken to identify predictors of haemoglobin concentration at baseline, at the time of initial cure assessment and at the time of assessment of definitive cure in a one-stage IPD meta-analysis. The regression models constructed at the time of initial and final cure assessments will adjust for the baseline measurements of haemoglobin as covariates along with the drug regimen.

If repeated haemoglobin measurements are available for more than three time points, the longitudinal haemoglobin profile will be characterised using a linear mixed-effects regression model by considering time as a continuous variable. Fractional polynomials will be used to explore any non-linear relationship in the evolution of haemoglobin concentration. If there are few numbers of time points, then time will be considered a discrete variable (three time points: baseline, day 30 and day 180) in the regression model. If frequent data is available, then the haemoglobin concentration will be summarised at weekly time points during the follow-up period.

The difference between haemoglobin concentration at baseline and at the time of evaluation of definitive cure status (usually at day 180) will be used as a proxy of absolute mean disease-attributed haematological fall. The difference between haemoglobin concentration at baseline and at the end of the active treatment phase (day 30) will be used to gauge if there is any drug-specific differences in haematological changes.

Further description of candidate predictors and multivariable modelling is described next.

### Candidate predictors and core set

The following variables will be considered for inclusion in the analysis of primary and secondary endpoints.

The following host variables are considered: age, gender, body weight, nutritional status, comorbidity status (such as HIV) and duration of illness prior to study enrolment. Nutritional status in children under 5 years of age will be assessed using standardised age, weight, height and gender-specific growth reference standards according to the WHO 2006 recommendations using the igrowup Stata package (or equivalent library will be used in R).[15]

The following parasite-related baseline factors will be considered: parasite grade and information regarding the nature of infection (primary vs previously treated cases). Any cases described as previously untreated (or 'fresh') cases for leishmaniasis will be considered as primary VL.

The following drug-related variables will be considered for inclusion in the analysis: drug regimen, mg/kg total dose (or target dose) and concomitant infections.

The following study or arm-level variables will be considered in the analysis of primary and secondary endpoints: geographical region, country, study site and calendar year of the study conduct.

The following covariates will be examined in the regression model and considered as core predictor sets: age, sex, baseline parasite density, HIV co-infection, geographical region and baseline haematological measurements. These variables, along with the drug regimen, will form the minimal adjustment set for the assessment of other risk factors and will be kept in the regression model regardless of statistical significance.[16]

### Considerations for multivariable model construction

Multivariable model construction will follow the recommendations of Heinze *et al* (2017).[17] Nested models will be compared by assessing the change in log-likelihood estimates and Akaike's information criterion will be used for comparing competing non-nested models. The functional form of the continuous variables will be determined using multivariable fractional polynomials[18] or restricted cubic splines.[16] Stability investigations will be undertaken to account for the uncertainty introduced in multivariable modelling through bootstrap resampling.[17]

### Handling missing data

To assess the impact of missing data, sensitivity analysis will be performed to see if the main conclusions are affected by the exclusion of patients with missing data using multiple imputation.[19] The imputation model will include all the variables in the target analysis and additional auxiliary variables. The target analysis will be carried out in each of the completed (observed plus imputed) datasets and the estimates will be combined across the imputed datasets using Rubin's combination rules.[19]

### Sensitivity analyses

Two sensitivity analyses will be carried out using resampling techniques to assess model stability. Bootstrap resampling will be used to assess the robustness of the derived estimates and their variance using the recommendations in Heinze *et al*.[17] In the second analysis, one study will be excluded at a time, and a coefficient of variation around the parameter estimates will be calculated.

### ANALYSIS OF SECONDARY ENDPOINTS
### Anaemia and severe anaemia at baseline, at the test of initial cure and at the end of the study follow-up

A mixed-effects logistic regression model will be constructed to identify the predictors associated with anaemia (or severe anaemia) at baseline using a one-stage IPD-MA. Random effects for the study sites will be used to adjust for the study-site effect.[20] Potential non-linear

relationships between continuous predictors and the outcome will be investigated using multivariable fractional polynomials.[21] Similar analysis will be undertaken for anaemia (or severe anaemia) at the time of initial cure and at the end of the study follow-up. Multivariable model construction will be undertaken as outlined for the primary endpoint.

### WBC and platelets (if data are available)

The distribution of WBC and platelets will be summarised at baseline, at the time of the initial cure assessment and at the time of the definitive cure assessment. If data are available, predictors associated with WBC (or platelet counts) will be carried out using a mixed-effects linear regression (appropriate transformation will be used if the distribution is skewed). Statistical modelling will be undertaken using mixed effects linear regression with multivariable modelling undertaken as for the primary endpoint.

### Blood transfusion during the study follow-up (if data are available)

Predictors associated with the requirement of blood transfusion at any stage during the study period will be identified (if data are available) using a mixed-effects logistic regression. Multivariable regression modelling will be undertaken as described for the other endpoints.

### Subgroup analyses

Patients living with HIV who are treated for VL typically have worse outcomes and a higher mortality risk than those without VL-HIV coinfections.[22] A separate subgroup analysis will be carried out among patients with defined VL-HIV coinfections (data permitting). There is a known regional variation in treatment response in VL, along with differences in patient characteristics and treatment guidelines.[23] Therefore, a separate analysis will be undertaken within each geographical region to construct the univariable and multivariable regression models for the primary and secondary outcomes.

The longitudinal haemoglobin profile will be stratified by the transfusion status at any stage of the study.

### Risk of bias assessment in included studies

To examine the risk of bias in IPD-MA, the first four domains of the quality in prognosis studies (QUIPS) tool and the first three domains of the prediction model risk of bias assessment tool (PROBAST) will be considered as recommended in Riley *et al*.[16] The relevant domains from the QUIPS checklist are study participation, study attrition, prognostic factor measurement and outcome measurement, and the relevant domains from the PROBAST checklist are participant selection, prognostic factors and outcomes. Two reviewers will independently assess the risk of bias in the studies included in the analysis.

Risk of bias results will be incorporated into analyses by conducting subgroup analyses among studies with an

overall low risk of bias or by conducting formal interaction analyses with a high risk of bias responses.[24]

## Assessment of risk of potential bias in missing studies

Despite the best possible efforts, it is anticipated that raw data from all the identified studies will not be available. The characteristics of the patient population and study metadata from the missing studies will be summarised to characterise if the missing studies are systematically different from the studies that are included in the IPD-MA. A two-stage IPD-MA will be considered if sufficient details (or any covariate adjustment) are reported in the original studies.

## Software

All the analysis will be carried out using R software or Stata V.17 software.[25 26] The use of any other data analysis tools will not change the statistical analysis plan.

## DISSEMINATION PLANS
### Ethics and dissemination

This IPD-MA meets the criteria for waiver of ethical review as defined by the Oxford Tropical Research Ethics Committee (OxTREC) granted to IDDO, as the research consists of secondary analysis of existing anonymised data. Ethics approval was granted by the ICMR-Rajendra Memorial Research Institute of Medical Sciences ethics committee (letter no.: RMRI/EC/30/2022) on 4 July 2022. Ethical approval was granted to each study included in this pooled analysis by their respective ethics committees. This IPD-MA will address research questions similar to those of the included studies. Findings from this IPD-MA will be reported in open-access, peer-reviewed journals following the PRISMA-IPD guidelines.[27]

### Patient and public involvement

The design and development of this IPD-MA were done by the study authors only and no patient was involved at any stage. Patients and/or the public were not involved in the design, or conduct, or reporting or dissemination plans of this research. The research questions considered in this IPD-MA are based on a research agenda developed by the global VL research community.[28]

### Further development of statistical analysis plan

The main analysis is planned as described above. Modifications or additional analyses may be required as the data collection progresses. Any modifications to the analysis will be documented and made publicly available on the IDDO study group website.[29]

## DISCUSSION

Anaemia is a common haematological manifestation of VL. Despite this, the impact of treatment on different haematological parameters remains to be fully understood. The aim of this IPD meta-analysis is to explore the dynamics of different haematological parameters during treatment and the convalescence phase of the disease. IPD-MA is being used increasingly to explore factors affecting treatment outcomes that otherwise would not be possible through standard aggregate data meta-analysis.[30]

This IPD-MA will provide critical information regarding the trajectory of haematological recovery among patients with VL. The assessment of the host, parasite and drug determinants that influence the haematological response can provide evidence-based guidance for optimal case management and monitoring drug safety. The IDDO VL library, which is a comprehensive library of all published studies since 1980, has been used for the identification of the studies eligible for this IPD-MA, which has led to the construction of the IDDO VL data platform. A major strength of this study is that data from several studies will be harmonised to a common standard (in the IDDO VL data platform) based on extensive consultation with the VL research community.[14]

A major challenge is that a substantial proportion of the studies in the IDDO library were conducted prior to the year 2000; the retrieval of data from historical trials is a major challenge.[31 32] Another limitation is that the exploration of the predictors of anaemia and severe anaemia at baseline will be influenced by the underlying eligibility criteria adopted in clinical trials, as clinical trials frequently exclude patients with very low haemoglobin at presentation.[33] Compared with non-pregnant patients, pregnant women are more at increased risk of requiring blood transfusion.[34] However, pregnancy is often adopted as an exclusion criterion in VL trials,[33] and the haematological consequences in this important patient group could not be explored despite the clear recognition of anaemia and transfusion needs.

This IPD-MA will characterise the haematological profile of patients with VL at enrolment and at the time points of initial and definitive cure assessments. The findings of this research will generate important information regarding the evolution of different haematological characteristics.

**Author affiliations**
[1]Infectious Diseases Data Observatory (IDDO), Oxford, UK
[2]Centre for Tropical Medicine and Global Health, Nuffield Department of Medicine, University of Oxford, Oxford, UK
[3]Rajendra Memorial Research Institute of Medical Sciences (RMRIMS), Patna, India
[4]Indian Council of Medical Research (ICMR), New Delhi, India
[5]Drugs for Neglected Disease Initiative, Geneva, Switzerland
[6]Infectious Disease Research Laboratory, Department of Medicine, Institute of Medical Sciences, Banaras Hindu University, Varanasi, India
[7]Médecins Sans Frontières, Amsterdam, Netherlands
[8]Institute of Endemic Diseases, University of Khartoum, Khartoum, Sudan

**Contributors** Study conception: AbM, PD, RK, SS-P, NAS, CN, JW, GB, MR, FA, PM, SS, KR, KS, KP, PJG and AhM. Project supervision: PD, AbM, KS and PJG. Methodology: PD, NAS, SS-P, JW, FA, PJG and KS. Data curation: AbM, SS-P, PM, JW, PD and GB. Project administration: SS-P and CN. Funding acquisition: PJG. Resources: SS-P, FA and PJG. Writing original draft: AbM, RK, SSP, KS, PJG, AhM and PD. Writing review and editing: all authors were involved in reading and critical revision of the initial draft and approved the final manuscript.

**Funding** This work is funded by a Bill & Melinda Gates Foundation grant to the Infectious Diseases Data Observatory, Oxford University, UK (recipient: Professor Philippe J. Guerin; ref: INV-004713). The funding agency had no role in developing the protocol.

**Competing interests** None declared.

**Patient and public involvement** Patients and/or the public were not involved in the design, or conduct, or reporting, or dissemination plans of this research.

**Patient consent for publication** Not applicable.

**Ethics approval** Not applicable.

**Provenance and peer review** Not commissioned; externally peer reviewed.

**ORCID iDs**
Abdalla Munir http://orcid.org/0000-0003-2994-0154
Prabin Dahal http://orcid.org/0000-0002-2158-846X
Sauman Singh-Phulgenda http://orcid.org/0000-0003-2892-3053
James Wilson http://orcid.org/0000-0003-3615-4928
Manju Rahi http://orcid.org/0000-0003-0932-0935
Krishna Pandey http://orcid.org/0000-0001-5930-2458

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
