## [Reviewer comments · BMJ Open]

ARTICLE DETAILS

TITLE (PROVISIONAL)	Haematological dynamics following treatment of visceral leishmaniasis: a protocol for systematic review and individual participant data (IPD) meta-analysis
AUTHORS	Munir, Abdalla; Dahal, Prabin; Kumar, Rishikesh; Singh-Phulgenda, Sauman; Siddiqui, Niyamat; Naylor, Caitlin; Wilson, James; Buck, Gemma; Rahi, Manju; Alves, Fabiana; Malaviya, Paritosh; Sundar, Shyam; Ritmeijer, Koert; Stepniewska, Kasia; Pandey, Krishna; Guérin, Philippe J.; Musa, Ahmed

VERSION 1 – REVIEW

REVIEWER	Cota, Gláucia Fundação Oswaldo Cruz (FIOCRUZ), Centro de Pesquisa René Rachou (CPqRR)
REVIEW RETURNED	03-Jun-2023

GENERAL COMMENTS	It is a new one very important initiative promoted by IDDO team, in this case aiming at mapping the dynamics of illness and recovery in VL, and mainly, the possible impact of different treatments on the hemantimetric parameter, a marker used in the diagnosis and assessment of cure. In general, the proposal is complete and it will use a robust and adequate statistical treatment. As an usual limitation of the combination of different studies, there are several biases that can be foreseen. Most of them have already been pointed out by the authors, considering the availability of information in the original publications. Here are just two suggestions to overcome the influence of the factors "the moment of quantification of the hematimetric parameters" and "blood transfusion", the last one capable to invert the direction of the effect, by correcting and masking the alteration. For the heterogeneity at the time of hemoglobin assessment in the follow-up, after starting treatment, I would suggest to work with shorter intervals (besides the single measurement between 15 and 60 days, as informed), measures grouped by week, even if the number of patients in each week is smaller, for an evaluation of the trend and speed of variation. It would provide a very useful information to be used in the stabliment of a more objective cure criteria. Regarding the spurious influence of transfusion, presenting the results stratified into two separeted groups, with or without transfusion in the treatment phase. Finally, I would suggest reconsider the purpose/statment of determining the prognostic markers of anemia at diagnosis, even though the intention is to correlate with major epidemiological and demographic variables. In a more cautious way, I would say it is about exploring factors possibly associated, most apropiate to the cross-sectional study design.
---

REVIEWER	Shafiei, Reza North Khorasan University of Medical Sciences
REVIEW RETURNED	21-Aug-2023

GENERAL COMMENTS	There will be many cases in the blood dynamics of affected patients. Environmental and hereditary factors are involved in this field. These factors can be effective during and after treatment. For example, the place of residence and the demographic and geographical factors of people's lives are important. On the other hand, in the laboratory protocols, the absence of parasites using molecular identification can help in the treatment. Considering the regional conditions and other epidemiological characteristics regarding VL, I cannot consider this model in the treatment of VL.
--

REVIEWER	Duminuco, Andrea University Hospital Gaspare Rodolico San Marco
REVIEW RETURNED	17-Sep-2023

GENERAL COMMENTS	This is a well-written and comprehensive study protocol regarding haematological parameters in visceral leishmania. An improvement in the evaluation of the patient's response to the treatment could be found in the evaluation of cytokine response, if available (consult DOI: 10.1016/j.biopha.2021.111671). More important, be careful about the cases of immunocompromised/haematological patients, where the response to treatment could be impaired (I suggest you refer to DOI: 10.3390/jcm12020578)
--

VERSION 1 – AUTHOR RESPONSE

Reviewer: 1
Dr. Gláucia Cota

Reviewer 1: Comments to the Author:

It is a very important initiative promoted by IDDO team, in this case aiming at mapping the dynamics of illness and recovery in VL, and mainly, the possible impact of different treatments on the hemantimetric parameter, a marker used in the diagnosis and assessment of cure. In general, the proposal is complete and it will use a robust and adequate statistical treatment.

Authors' response: We would like to thank Dr. Cota for the kind comments on our protocol.

Reviewer 1: As a usual limitation of the combination of different studies, there are several biases that can be foreseen. Most of them have already been pointed out by the authors, considering the availability of information in the original publications. Here are just two suggestions to overcome the influence of the factors "the moment of quantification of the hematimetric parameters" and "blood transfusion", the last one capable to invert the direction of the effect, by correcting and masking the alteration.

For the heterogeneity at the time of haemoglobin assessment in the follow-up, after starting treatment, I would suggest to work with shorter intervals (besides the single measurement between 15 and 60

days, as informed), measures grouped by week, even if the number of patients in each week is smaller, for an evaluation of the trend and speed of variation. It would provide a very useful information to be used in the establishment of a more objective cure criteria.

Authors' response: Thank you for this suggestion regarding the time-point of measurement of haematological concentration. The following is added to the manuscript.

Lines 228-230: If frequent data is available, then the haemoglobin concentration will be summarised at weekly time-points during the follow-up period.

Reviewer 1: Regarding the spurious influence of transfusion, presenting the results stratified into two separated groups, with or without transfusion in the treatment phase.

Authors' response: Thank you for this. The following is now added in the lines 331-332:

Lines 317-318: Longitudinal haemoglobin profile will be stratified by the transfusion status at any stage of the study.

Reviewer 1: Finally, I would suggest reconsider the purpose/statement of determining the prognostic markers of anaemia at diagnosis, even though the intention is to correlate with major epidemiological and demographic variables. In a more cautious way, I would say it is about exploring factors possibly associated, most appropriate to the cross-sectional study design.

Authors' response: We agree with this limitation and this is now acknowledged in the discussion:

Lines 377-380:

Another limitation is that the exploration of the predictors of anaemia and severe anaemia at baseline will be influenced by the underlying eligibility criteria adopted in clinical trials as clinical trials frequently exclude patients with very low haemoglobin at presentation.³³

Reviewer: 2

Dr. Reza Shafiei, North Khorasan University of Medical Sciences

Reviewer 2: There will be many cases in the blood dynamics of affected patients. Environmental and hereditary factors are involved in this field. These factors can be effective during and after treatment. For example, the place of residence and the demographic and geographical factors of people's lives are important.

Authors' response: Thank you for this comment. We have tried to consider some of the factors suggested such as place of residence (by taking geographical region into account when modelling; please see lines: lines 255-259 and lines 314-316 for a detailed description on these).

Several demographic characteristics are considered in our analysis (age, sex, nutritional status). However, we are unable to account for hereditary factors as such data is not typically collected in clinical trials settings.

Reviewer 2: On the other hand, in the laboratory protocols, the absence of parasites using molecular identification can help in the treatment.

Authors' response: While we haven't considered this explicitly in our protocol, we agree with the reviewer that variability in lab methodology adopted for judging parasitological presence (eg: molecular methods such as PCR or microscopy or serological) remains important. Such variability in underlying methodology for parasitological assessment will be described in the trial characteristics and will form part of the sensitivity analysis. A slight modification to acknowledge the possible impact of such methodological variation in defining outcomes in lines 283-286 in the sensitivity analysis section:

Lines 283-286: This would identify any influential studies, that is, studies with unusual results (due to variations in underlying methodology to measure outcomes, patient population, and so on) that affect the overall pooled analysis findings.

Reviewer 2: Considering the regional conditions and other epidemiological characteristics regarding VL, I cannot consider this model in the treatment of VL.

Authors' response: As stated in our earlier response, we have considered incorporation of region and underlying patient demographics in our analysis.

Lines 313-316: There is a known regional variation in treatment response in VL along with differences in patient characteristics and treatment guidelines.²³ Therefore, a separate analysis will be undertaken within each geographical region to construct the univariable and multivariable regression models for the primary and secondary outcomes.

Reviewer: 3

Dr. Andrea Duminuco, University Hospital Gaspare Rodolico San Marco

Reviewer 3: This is a well-written and comprehensive study protocol regarding haematological parameters in visceral leishmania. An improvement in the evaluation of the patient's response to the treatment could be found in the evaluation of cytokine response, if available (consult DOI: 10.1016/j.biopha.2021.111671). More important, be careful about the cases of immunocompromised/haematological patients, where the response to treatment could be impaired (I suggest you refer to DOI: 10.3390/jcm12020578)

Authors' response: Thank you for these helpful references pointed out to us. These will be a very useful reference when interpreting the findings of our IPD-MA.